# Genetic and environmental influence on alcohol intent and alcohol sips among U.S. children–Effects across sex, race, and ethnicity

**Troy Puga** [1,2]*, **Yadi Liu** [1], **Peng Xiao** [3], **Ran Dai** [1], **Hongying Daisy Dai** [1]*

**1** College of Public Health, University of Nebraska Medical Center, Omaha, NE, United States of America,
**2** College of Osteopathic Medicine, Kansas City University, Kanas City, MO, United States of America,
**3** Dept. of Genetics, Cell Biology & Anatomy, University of Nebraska Medical Center, Omaha, NE, United States of America

* troy.puga@kansascity.edu (TP); daisy.dai@unmc.edu (HDD)

**Data Availability Statement:** All data used in this study is publicly available in the ABCD Database in the National Data Archive at the following DOI:10.15154/z563-zd24.

## Abstract

### Introduction

Alcohol intent (the susceptibility to initiating alcohol use) and alcohol sips (the initiation of alcohol) in youth are a multifactorial puzzle with many components. This research aims to examine the connection between genetic and environmental factors across sex, race and ethnicity.

### Methods

Data was obtained from the twin hub of the Adolescent Brain Cognitive Development (ABCD) study at baseline (2016–2018). Variance component models were conducted to dissect the additive genetic (A), common (C) and unique environmental (E) effects on alcohol traits. The proportion of the total alcohol phenotypic variation attributable to additive genetic factors is reported as heritability ($h^2$).

### Results

The sample (n = 1,772) included an approximately equal male-female distribution. The 886 same-sex twin pairs were 60.4% dizygotic (DZ), 39.6% monozygotic (MZ), 65.4% non-Hispanic Whites, 13.9% non-Hispanic Blacks, 10.8% of Hispanics with a mean age of 121.2 months. Overall, genetic predisposition was moderate for alcohol intent ($h^2 = 28\%$, $p = .006$) and low for alcohol initiation ($h^2 = 4\%$, $p = 0.83$). Hispanics ($h^2 = 53\%$, $p < .0001$) and Blacks ($h^2 = 48\%$, $p < .0001$) demonstrated higher alcohol intent due to additive genetic factors than Whites ($h^2 = 34\%$, $p < .0001$). Common environmental factors explained more variation in alcohol sips in females ($c^2 = 63\%$, $p = .001$) than in males ($c^2 = 55\%$, $p = .003$). Unique environmental factors largely attributed to alcohol intent, while common environmental factors explained the substantial variation in alcohol initiation.

### Conclusion

Sex and racial/ethnic disparities in genetic and environmental risk factors for susceptibility to alcohol initiation can lead to significant health disparities. Certain populations may

**Funding:** Research by H. Dai was partially supported by the National Institute on Drug Abuse under Award Numbers R21DA054818 and R01DA058992. The funders had no role in study design, data collection and analysis, decision to publish, or preparation of the manuscript.

**Competing interests:** The authors have declared that no competing interests exist.

be at greater risk for alcohol use due to their genetic and ecological factors at an early age.

## Introduction

Alcohol initiation and use is a complex problem that is often multifactorial. Genetic and environmental factors are two components that may offer insights into the vulnerability of adolescents to alcohol susceptibility and use. The current levels of adolescent alcohol use have declined in recent years [1], particularly through the COVID-19 Pandemic [2, 3]. Binge drinking among youth has decreased as well [3]. However, there remains a high number of adolescents who binge drink [4]. While the overall alcohol use levels in youth have decreased, there remain disparities in factors such as sex, race, and ethnicity.

While the percentage of boys (19%) who start drinking before the age of 14 was higher than that of girls (13%) [5, 6], those girls who start drinking during early adolescence experience a shorter time span between their first drink and their first occurrence of binge drinking [7]. Young males may be more likely to engage in alcohol-drinking behavior than females due to the current peer pressures in their social circles and a strong desire to demonstrate masculinity amongst their peers [8, 9].

Research has shown that African Americans and Hispanics are more likely to develop drinking consequences or symptoms of dependence than Whites [10]. In African Americans, lower socioeconomic status and weaker school bonds were correlated with an increased risk of alcohol use disorder [11]. However, African Americans are likely to initiate alcohol-engaging behaviors later than whites [12, 13]. Current research has shown that Hispanics tend to initiate alcohol sooner than whites [14, 15] and may have a strong genetic predisposition for alcohol initiation [16, 17]. These racial and sex differences are important factors that need to be further explored regarding adolescent alcohol use. Moreover, it is essential to investigate factors like the social determinants of health, which have been significantly implicated in alcohol usage and addiction [18]. Currently, there is limited research conducted to investigate the interplay between genetics, environmental factors, race, ethnicity, and sex regarding alcohol use among adolescents in the United States.

Genetic factors play a significant role in the initiation and usage of substances. Research has shown that alcohol addiction has been linked to several genetic factors. Certain genes can predispose individuals to alcohol addiction, while other genes may confer protection against alcohol addiction [16, 17]. Previous twin studies have shown that alcohol dependence is heritable [19, 20]. It is worth noting that these findings are predominantly derived from studies involving adult participants. Twins are vital to studying heritable factors on substance use in adolescents due to their similar genetic components. Monozygotic twins, known as identical twins [21], come from the same egg, which splits into two eggs early in the development process [21]. These twins share the same genetic DNA but may be influenced by epigenetics, which provides some small variations [21, 22]. Dizygotic twins come from two separate eggs but share 50% of segregating genes. Twin studies can offer a valuable approach for behavioral genetics to assess the interplay of genes and the environments on complex traits.

Currently, several studies suggest that genetic predisposition may contribute to alcohol intent (the susceptibility to initiating alcohol use), alcohol sips (the initiation of alcohol), or the development of alcohol use disorder. A previous meta-analysis showed that alcohol use disorder is approximately 50% heritable [19], and additional genomic analyses were conducted to

further examine what specific portion of the genome is responsible for this [23]. However, it is believed that the predisposition to alcohol use is influenced by the interaction of many genes [23, 24], with one study identifying over 200 genes involved in the predisposed risk of alcohol use disorder [25]. Limited research exists on conducting specific gene investigations with respect to sex, race, and ethnicity [26, 27].

The development of children is significantly influenced by social environments, and environmental factors have been linked to the onset of numerous chronic diseases and mental health conditions [28–30]. Factors such as neighborhood safety, education, and health may play a significant component in the intent and sips of alcohol usage and addiction [31, 32]. Addiction has been linked to social environments and the chronic stressors that individuals may experience [33], and alcohol could function as both a substance and a coping mechanism in response to stress arising from their social environment [34].

The complex interplay between genetic and environmental factors in both alcohol intent and sips can lead to variations in associated risk factors related to sex, race, and ethnicity during the early stage of life. In this behavioral genetic study, we aim to assess the extent to which additive genetic and environmental influences affect early-age susceptibility to alcohol and initiation, particularly among sex, racial, and ethnic subgroups. To do so, we evaluated and compared monozygotic (MZ) and dizygotic (DZ) twins using data from the Adolescent Brain and Cognitive Development (ABCD) Study. The goal of this study is to explore three inquiries among children aged 9 and 10 years old: 1) How do genetic and environmental factors impact youth alcohol use behaviors? 2) Is there a difference in the genetic predisposition of alcohol use between males and females? 3) To what extent do genetic predisposition and environmental influences vary among individuals of different racial and ethnic backgrounds?

## Methods

### Data

The ABCD study is the largest study in the United States in the area of brain development and child health. A total of 11,880 children aged 9 and 10 years were enrolled at baseline at 21 U.S. study sites [35]. Participants were recruited through a probability sample of schools selected for sex at birth, race/ethnicity, socioeconomic status, and urbanicity to maintain the sample demographics in accordance with the American Community Survey 3rd and 4th-grade enrollment statistics at each site [36]. This study has collected data from the ABCD study Twin Hub, with same-sex twins enrolled at baseline from 4 leading twin research centers (Washington University in St. Louis, University of Minnesota, University of Colorado at Boulder, and Virginia Commonwealth University). An equal number of twin pairs born during 2006–2008 were recruited from registries in each state [37]. Study procedures were approved by the University of California San Diego Central Institutional Review Board (IRB) and each local institutional IRB for the ABCD study. The Institutional Review Board from the University of Nebraska Medical Center granted an exemption for the secondary data analysis of de-identified information from the ABCD study.

Zygosity was extrapolated from the ABCD dataset as described below. Outliers, including those of uncertain zygosity after characterization or who were found to lack a corresponding twin, were excluded from the dataset.

### Measures

**Zygosity status.** Participants' family relationships were obtained from parents who were given response options such as "Single," "Sibling," "Twin," and "Triplet." For twins, genetic inference was initially used to determine their zygosity status, categorizing them as either MZ or DZ. In

instances where zygosity status was unavailable, researchers resorted to physical characteristics, such as facial appearance, complexion, hair features (color, texture, curliness, pattern, amount, and darkness), ear appearance, hair type, and eye color, to differentiate between MZ and DZ twins [37]. Of 1,069 twin pairs (2,138 participants), we excluded 169 pairs (n = 338) with missing zygosity status, 12 pairs (n = 24) with inconsistent zygosity, and 2 pairs (n = 4) with families including only one individual. Ultimately, 886 pairs of twins (n = 1,772) were utilized for the present study.

**Alcohol intent and sips.** Participants were first asked whether they had heard of alcohol products, such as beer, wine, or liquor. Those who reported "Yes" were asked whether they had ever tried a sip of alcohol such as beer, wine, or liquor (rum, vodka, gin, whiskey) at any time in their life. Those who reported "Yes" were classified as ever alcohol users (alcohol initiation) [38].

Never alcohol users were further asked, "Have you ever been curious about drinking alcohol?" with response options of "Very curious," "Somewhat curious," "A little curious," and "Not at all curious." They were also asked "Do you think you will try alcohol soon?" and "If one of your best friends were to offer you alcohol, would you try it?" with response options "Definitely yes," "Probably yes," "Probably not," and "Definitely not." Those who reported "Not at all curious" and "Definitely not" to all three susceptibility questions were classified as "not susceptible to alcohol use" [39, 40].

**Sociodemographic factors.** Sociodemographic factors, which were provided by parents, were used to describe sample characteristics of the participants. These variables included age (measured continuously in months), sex assigned at birth (male/female), race/ethnicity (including Non-Hispanic White, Non-Hispanic Black, Hispanic, and other Non-Hispanic races), parent's highest education level (categorized as less than high school, high school, some college or associate degree, Bachelor's degree, or postgraduate degree), family income (grouped as less than $25,000, $25,000-$49,999, $50,000-$74,999, $75,000-$99,999, $100,000+, or don't know/refuse to answer), family difficulty experienced in the past 12 months (Yes/No), and premature birth status (Yes/No). Measures of perceived neighborhood safety and crime were also used, which were reported by both the youth and their parents, with higher scores indicating a safer perception of the neighborhood [41].

## Statistical methods

Sociodemographic information was collected and compiled. Rao-Scott Chi-Square tests were conducted to detect sociodemographic differences between the intent and sip groups. See Table 1. The classical twin model estimates three sources of variance: additive genetic (A), shared common environmental (C), and unique environmental (E) factors. Each source of variance is latent and estimated from the similarity in the correlations of twin pairs on a phenotype. We constructed an ACE analysis using a mixed model $y_{ij} = X_{ij}\beta + a_{ij} + c_i + \epsilon_{ij}$, where $i$ is the index for $n_1$ MZ and $n_2$ DZ twin pairs [42, 43], and $a$, $c$ measure the additive genetic, and common environmental random effects on the $i^{th}$ twin pair. The magnitude of difference in the correlation of a particular phenotype by zygosity is used to attribute additive genetic or shared environmental sources of variance in various scenarios. For instance, 1) when the MZ correlation is larger than (e.g., 50%) the DZ correlation (30%), there is evidence of genetic influences, 2) when the MZ correlation (e.g., 42%) is similar to the DZ correlation (e.g., 42%), variance is likely attributed to environmental factors with little evidence of genetic influence, and 3) when the DZ correlation (60%) is larger than half of the MZ correlation (80%), variance is likely attributed to environmental factors [20, 44]. Heritability ($h^2$, *the proportion of the total phenotypic variation attributable to genes*), and percentage of variances from shared ($c^2$) and unique

**Table 1. Sample characteristics and prevalences of alcohol traits among twins aged 9 and 10 years, 2016–2018.**

| Sociodemographics | n (%) | Intent | P-value[b] | Sips | P-value[a] |
|---|---|---|---|---|---|
| **All** | **1772 (100%)** | **23.5 (21.1–25.9)** | | **25.4 (23.4–27.4)** | |
| **Age, mean (SD), months** | **121.2(6.7)** | | | | |
| Sex at birth[b] | | | 0.003 | | 0.003 |
| Male | 905(51.1%) | 27.2 (23.6–30.7) | | 28.4 (25.5–31.3) | |
| Female | 867(48.9%) | 19.9 (16.8–23.1) | | 22.3 (19.5–25.0) | |
| Race/ethnicity[b] | | | 0.46 | | < .0001 |
| White | 1159(65.4%) | 24.0 (21.0–27.0) | | 28.6 (26.0–31.3) | |
| Black | 246(13.9%) | 19.0 (13.4–24.7) | | 8.1 (4.7–11.5) | |
| Hispanic | 192(10.8%) | 25.4 (17.9–32.9) | | 24.0 (17.9–30.0) | |
| Other | 174(9.8%) | 25.0 (17.1–32.9) | | 29.3 (22.5–36.1) | |
| Parental education level[b] | | | 0.15 | | < .0001 |
| Less than high school | 43(2.4%) | 12.5 (1.0–24.0) | | 11.6 (2.0–21.2) | |
| High school diploma | 109(6.2%) | 14.8 (7.1–22.6) | | 12.8 (6.6–19.1) | |
| Some college | 436(24.6%) | 22.9 (18.3–27.6) | | 20.4 (16.6–24.2) | |
| Bachelor degree | 618(34.9%) | 25.5 (21.3–29.6) | | 25.6 (22.1–29.0) | |
| Postgraduate degree | 566(31.9%) | 24.5 (20.0–29.0) | | 32.5 (28.6–36.4) | |
| Family income[b] | | | 0.03 | | 0.0005 |
| <$25,000 | 126(7.1%) | 22.2 (13.6–30.8) | | 15.1 (8.8–21.3) | |
| $25,000-$49,999 | 160(9.0%) | 18.5 (11.5–25.5) | | 17.5 (11.6–23.4) | |
| $50,000-$74,999 | 240(13.5%) | 15.1 (9.8–20.5) | | 22.1 (16.8–27.3) | |
| $75,000-$99,999 | 251(14.2%) | 26.0 (19.5–32.5) | | 24.3 (19.0–29.6) | |
| $100,000+ | 893(50.4%) | 26.6 (23.0–30.2) | | 29.6 (26.6–32.6) | |
| Don't know or refuse to answer | 102 (5.8%) | 21.2 (11.3–31.1) | | 24.5 (16.2–32.9) | |
| Family Difficulty[b,c] | | | 0.80 | | 0.62 |
| No | 1596(90.1%) | 23.6 (21.1–26.1) | | 25.6 (23.4–27.7) | |
| Yes | 176(9.9%) | 22.5 (14.7–30.3) | | 23.9 (17.6–30.2) | |
| Premature[b] | | | 0.36 | | 0.13 |
| No | 763 (43.4%) | 24.7 (21.0–28.3) | | 23.7 (20.7–26.7) | |
| Yes | 994 (56.6%) | 22.4 (19.2–25.6) | | 26.9 (24.1–29.6) | |
| Neighborhood perceptions[d], Mean (SE) | | | | | |
| Child | 4.2(1.0) | | | | |
| Parent | 4.2(0.8) | | | | |

[a] Rao-Scott Chi-square tests were performed to detect group differences.

[b] n (column %)

[c] Experience of any family difficulty in the past 12 months was assessed by 7 items, e.g., "need food but couldn't afford it," "didn't pay the full amount of the rent or mortgage because you could not afford it" (Cronbach's alpha = .91).

[d] Perceived neighborhood safety and crime assessed feelings about safety and the presence of crime in the respondent's neighborhood, including measures from the youth (one item with a 5-point Likert scale, "My neighborhood is safe from crime") as well as the parents (average of three items with a 5-point Likert scale, "I feel safe walking in my neighborhood, day or night." Violence is not a problem in my neighborhood." and "My neighborhood is safe from crime." Cronbach's alpha = .89) Higher scores indicated a safer perceived neighborhood (1-Strongly Disagree, 5-Strongly Agree).

($e^2$) environments were reported. Intraclass correlation coefficients, alongside Pearson correlation coefficients and Bayesian information criterion (BIC) were provided to inform model fit [45]. Due to the identifiability problem, we did not dissect additive genetic effects, dominance genetic effects, and shared environmental effects. Statistical analyses were performed using SAS with 95% confidence level (two-tailed p-value<0.05) [42].

The statistical power for the twin study was performed by Iacona *et al*. [37]. With 800 twin pairs recruited from four geographically diverse sites (Minnesota, Colorado, Virginia, and Missouri), the ABCD study offers the largest sample size to assess behavioral genetics among adolescents longitudinally. Because DZ twins are twice more prevalent than MZ pairs, the ABCD twin study oversampled MZ twins to balance the zygosity distribution. The power analysis considered the impact of sample size and measurement type on detecting heritable variation. With the proposed sample, this study achieved a power greater than 0.8 to detect additive genetic effects when the heritable variation exceeds 0.3 for continuous measures. The power of binary measures depends on the prevalence of outcomes, while ordinal measures provide more power than binary measures. Despite practical limitations, the study was designed to maximize statistical power. See Iacona *et al*. [37] for detailed power analysis results.

## Results

The analytical sample used in the study consisted of 886 twin pairs with a mean (standard deviation) age of 121.2[6.7] months. The sample had a similar distribution of male (51.1%) and female (48.9%) participants and included 65.4% Whites, 13.9% Blacks, and 10.8% Hispanics. Overall, 23.5% of never-alcohol users reported intent to use alcohol, and 25.4% of participants reported alcohol sips at the age of 9–10 years old. There was a statistically significant difference between males and females regarding intent (p = .003) and sips (p = .003). Males had a higher prevalence of intent and sips of alcohol use than females. Alcohol sips differed by race and ethnicity (p < .0001), with Blacks reporting the lowest prevalence of alcohol sips (8.1% vs. 28.6% for non-Hispanic Whites). There was a significant difference in alcohol sips (p < .0001) by parental education level and family income. Children with higher parental education (p < .0001) and higher family income (p = .0005) tended to have a higher prevalence of alcohol sips. See Table 1.

Environmental and genetic factors that influenced both the intent and sips of alcohol are presented in Table 2. The susceptibility to alcohol use was correlated within monozygotic twins (r = 0.19, p = .007) but not within dizygotic twins, leading to variation attributable to heritability ($h^2$ = 28%, p = .006) and unique environment ($e^2$ = 72%, p < .0001). Early-age alcohol sips were correlated within both dizygotic (r = 0.44, p< .0001) and monozygotic twins (r = 0.44, p < .0001) with a similar magnitude. The analysis of the ACE model revealed that the common environment ($c^2$ = 62%, p < .0001) and unique environment ($e^2$ = 35%, p < .0001) are the primary factors affecting alcohol sips, while additive genes appear to have an insignificant influence on alcohol sips at the age of 9–10 years old ($h^2$ = 4%, p = 0.83).

Genetic predisposition and environmental influences of alcohol intent by sex and race/ethnicity are presented in Table 3. Intent was significantly correlated within MZ twins in males (r = 0.22, p = .03) and Hispanics (r = 0.44, p = .03). As compared to females, males showed higher intent to alcohol initiation (27.2% vs. 19.9%). Hispanics ($h^2$ = 53%, p < .0001) and Blacks ($h^2$ = 48%, p < .0001) demonstrated higher intent to alcohol initiation due to genetic predisposition than Whites ($h^2$ = 34%, p < .0001).

Alcohol sips were correlated in MZ and DZ twins by sex, race, and ethnicity (r = 0.38–0.66, p < .0001), except in Black DZ twins (r = 0.05, p = .69). The ACE analysis reported a common environment as the largest source of variance ($c^2$ = 55%, p = .003 for males; $c^2$ = 63%, p = .001 for females), followed by a unique environment ($e^2$ = 36%, p < .0001 for males and $e^2$ = 34%, p = .0007 for females). See Table 4.

Among racial and ethnic groups, the ACE model demonstrated that a common environment accounted for the largest variance in Whites ($c^2$ = 60%, p < .0001) and Hispanics ($c^2$ = 54%, p = .02). Addictive genes accounted for the largest variance in Blacks ($h^2$ = 54%) though they are not statistically significant (p = .13). See Table 4.

**Table 2. Variance component analysis of genetic heritability for alcohol-related traits.**

|  | Alcohol Intent | Alcohol Sips |
|---|---|---|
| **Prevalence (%)** |  |  |
| All (n = 1772) | 23.5 | 25.4 |
| Dizygotic (n = 1070, 60.4%) | 24.0 | 25.1 |
| Monozygotic (n = 702, 39.6%) | 22.6 | 25.9 |
| **Pearson Correlation (p-value)** |  |  |
| Dizygotic Twins | r = 0.05 (p = 0.37) | r = 0.44 (p < .0001) |
| Monozygotic Twins | r = 0.19 (p = 0.007) | r = 0.44 (p < .0001) |
| **Intraclass Correlation Coefficient (ICC)** |  |  |
| Dizygotic Twins | ICC = 0.06 | ICC = 0.43 |
| Monozygotic Twins | ICC = 0.20 | ICC = 0.44 |
| **ACE model [a]** |  |  |
| Additive Genes ($h^2$) | 0.28 (0.08 to 0.49) [b] P = 0.006 [b] | 0.04 (-0.30 to 0.37) p = 0.83 |
| Common Environment ($c^2$) | 0 | 0.62 (0.37 to 0.86) P < .0001 |
| Unique Environment ($e^2$) | 0.72 (0.52 to 0.92) [b] P < .0001 [b] | 0.35 (0.22 to 0.47) P < .0001 |
| BIC | 1331.8 [b] | 1883.7 |

[a]Variance component methods (e.g., structural equation models) were performed using SAS Proc Mixed and Proc NLMixed. Multiple variance decompositions methods, including ACE, AE, and CE, were compared using BIC, model goodness-of-fit statistics. See Supplementary Material 2 in S1 Appendix.

[b] Since common envioronoment $c^2$ = 0, we reported the AE model result for alcohol intent after model comparison and reduction. See details of ACE and AE models in Supplementary e Table 2 in S1 Appendix.

**Table 3. Estimates of genetic and environmental variance components for intent of alcohol use, stratified by sex and race/ethnicity.**

|  | Male | Female | White | Black | Hispanic |
|---|---|---|---|---|---|
| **Prevalence (%)** |  |  |  |  |  |
| All | 27.2 | 19.9 | 24.0 | 19.0 | 25.4 |
| DZ | 28.2 | 20.3 | 26.0 | 16.8 | 26.6 |
| MZ | 25.6 | 19.4 | 21.0 | 23.4 | 23.5 |
| **Correlation** |  |  |  |  |  |
| DZ | r = 0.02 p = 0.8 | r = 0.07 p = 0.39 | r = 0.09 p = 0.25 | r = -0.08 p = 0.57 | r = 0.01 p = 0.95 |
| MZ | r = 0.22 p = 0.03 | r = 0.15 p = 0.16 | r = 0.06 p = 0.50 | r = 0.32 p = 0.11 | r = 0.44 p = 0.03 |
| **ACE model [a]** |  |  |  |  |  |
| Additive Genes ($h^2$) | 0.38 (0.31 to 0.44) p < .0001 | 0.39 (0.067 to 0.71) p = 0.02 | 0.34 (0.29 to 0.40) p < .0001 | 0.48 (0.44 to 0.52) p < .0001 | 0.53 (0.39 to 0.67) p < .0001 |
| Unique Environment ($e^2$) | 0.62 (0.56 to 0.69) p < .0001 | 0.61 (0.29 to 0.93) P = 0.0002 | 0.66 (0.60 to 0.71) p < .0001 | 0.52 (0.48 to 0.56) p < .0001 | 0.47 (0.33 to 0.61) p < .0001 |
| BIC | 99.2 | 77.9 | 222.1 | 65.4 | 62.1 |

[a]: The estimate for common environment ($c^2$) is 0, as indicated in Table 1.

**Table 4. Estimates of genetic and environmental variance components for sips of alcohol use, stratified by sex and race/ethnicity.**

| | Male | Female | White | Black | Hispanic |
|---|---|---|---|---|---|
| **Prevalence (%)** | | | | | |
| All | 28.4 | 22.3 | 28.6 | 8.1 | 24.0 |
| DZ | 28.0 | 22.1 | 29.2 | 4.4 | 22.0 |
| MZ | 29.1 | 22.5 | 27.9 | 14.8 | 27.0 |
| **Correlation** | | | | | |
| DZ | r = 0.42 p < .0001 | r = 0.44 p < .0001 | r = 0.41 p < .0001 | r = -0.05 p = 0.69 | r = 0.51 p < .0001 |
| MZ | r = 0.43 p < .0001 | r = 0.43 < .0001 | r = 0.38 p < .0001 | r = 0.55 p = 0.0001 | r = 0.66 p < .0001 |
| **ACE model** | | | | | |
| Additive Genes ($h^2$) | 0.09 (-0.38 to 0.56) p = 0.70 | 0.03 (-0.50 to 0.55) p = 0.91 | 0 | 0.54 (-0.16 to 1.24) p = 0.13 | 0.32 (-0.20 to 0.84) p = 0.23 |
| Common Environment ($c^2$) | 0.55 (0.19 to 0.91) p = 0.003 | 0.63 (0.25 to 1.01) p = 0.001 | 0.60 (0.50 to 0.70) p < .0001 | 0.21 (-0.73 to 1.15) p = 0.66 | 0.54 (0.077 to 1.00) p = 0.02 |
| Unique Environment ($e^2$) | 0.36 (0.19 to 0.52) p < .0001 | 0.34 (0.14 to 0.54) p = 0.0007 | 0.40 (0.30 to 0.50) p < .0001 | 0.25 (-0.19 to 0.69) p = 0.27 | 0.14 (-0.052 to 0.34) p = 0.15 |
| BIC | 1035.4 | 876.8 | 1328.0 | 147.5 | 202.9 |

## Discussion

By analyzing a large number of DZ and MZ twin pairs, this study found moderate genetic predisposition with alcohol intent and low genetic predisposition in early-age alcohol sips. Children aged 9 to 10 typically experience significant physical, cognitive, and social-emotional development [46]. Middle childhood is a crucial developmental stage in which children establish their identities and form social connections with peers, which can influence their attitudes and behaviors toward alcohol use [47, 48]. The findings from this study indicate that, at the early stage of life, genetic factors play a relatively small role in determining whether a child will try alcohol for the first time at that age. Instead, environmental factors such as peer pressure, social norms, and family environment are likely to have a greater influence on a child's decision to initiate alcohol use.

The stratified analysis by race and ethnicity indicates that Hispanics might have a higher genetic component for alcohol intent. Other studies have found that Hispanic children may be more likely than White children to engage in binge drinking [14, 15]. We found that alcohol susceptibility among Hispanic children was primarily explained by genetic and unique environmental factors. For Hispanic youth, genetic predisposition plays a significant role in determining the likelihood of an individual engaging in alcohol use during adolescence [19, 20]. Additionally, unique environmental factors such as peer influence or marketing exposure can shape individual alcohol intent.

This study also adds to the literature by finding that Black youth might have a higher genetic predisposition for alcohol sips. We found that Black children had a lower prevalence of alcohol sips than their White counterparts. A strong sense of cultural identity, a positive self-concept, and pro-social behavior [49, 50] may protect Black youth from initiating alcohol, as shown by the low unique environmental influence on alcohol sips among Black children. These findings are consistent with literature indicating that black youth adolescents initiate alcohol use at later ages than their White peers and that they are less likely to engage in binge drinking than White youth [12, 13]. Collectively, we postulate that genetic factors are more likely to influence alcohol sips among Black youth at a later age.

In our study, males showed a higher prevalence of alcohol sips and greater influence from common environmental factors than females. For both males and females, alcohol intent was primarily influenced by a unique environment and followed by genetic liability, while common and unique environments determined alcohol initiation. Genetic and environmental factors might influence boys and girls differently during adolescence. For instance, boys might undergo socialization that encourages a greater acceptance of risk-taking behaviors and alcohol use as a means to demonstrate masculinity [8]. Additionally, peer pressure and social norms may also influence adolescent drinking behaviors, as boys and girls may experience different expectations and pressures from their social groups [9].

This study found that genetic factors are associated with alcohol intent. As the study population comprised 9–10 years of children, alcohol use behaviors might be in the early phase with children starting to explore risk behaviors. Thus, genetic effects on alcohol sips might play a more prominent role in the later stage of life. Previous research has shown genetic and environmental factors contribute to chronic diseases and substance use disorders [16, 28–30]. Studies have identified variants in genes such as alcohol dehydrogenase (ADH) or aldehyde dehydrogenase (ALDH) that can either be protective from alcohol use disorder or a potential risk [24, 51]. Genes such as ADH1B and ALDH2 have been shown to be protective in Asian populations, while ADH1C has been shown to increase the risk for alcohol use disorder [51]. Future studies are necessary to identify specific genes that are linked to alcohol use among Black and Hispanic populations [25].

The findings in this study will be valuable for informing future public health interventions in reducing early-age alcohol initiation and alcohol use disorder. The knowledge gained from this study will allow public health practitioners to apply health promotion theories and interventions to individuals who may be at higher risk for alcohol intent and initiation. Public health interventions that use the information from this study will have the potential to reduce early-age alcohol use in the high-risk populations.

We acknowledge that this study had some limitations. First, children are prone to social desirability bias when reporting substance use, including alcohol intent and sips, especially for younger adolescents [52]. Second, the study did not specify the environmental factors that may increase the risk of early-age alcohol susceptibility and initiation beyond unique and shared influences. Third, the study was limited regarding the time since alcohol sips, as it could have begun anytime in early childhood. It would be beneficial for future research to investigate more specific environmental factors, such as exposure to childhood adverse experiences, parental monitoring, school environment, and neighborhood characteristics, that may contribute to the elevated risk of alcohol use in the future. Fourth, this study excluded cases where there were errors or inconsistencies in the recorded zygosity information due to concerns about the accuracy of data entry. Future research should investigate using raw maximum likelihood estimates in structural equation modeling, as this method can accommodate incomplete data [53]. Finally, while the ABCD study has one of the largest twin hub data for longitudinal assessment of behavioral genetics among adolescents, its capacity to detect racial disparities might be limited in the ACE model due to the relatively small sample sizes within certain racial minority groups (e.g., Blacks ad Hispanics). Therefore, caution should be exercised when generalizing findings to these populations, and additional research with larger samples from diverse racial backgrounds may be necessary to fully understand the implications for different demographic groups.

## Conclusion

Despite these limitations, this study showed that genetic and environmental factors are correlated with both alcohol intent and sips. Overall genetic predisposition was moderate for

alcohol intent and low for alcohol sips, but the effects from genetic and environmental factors varied by sex and racial/ethnic groups. Disparities in genetic and environmental risk factors related to sex and race/ethnicity can increase the vulnerability of certain populations to alcohol use and associated adverse health outcomes. This highlights the need to address these disparities and develop targeted interventions to prevent early-onset alcohol use in at-risk populations.

## Supporting information

**S1 Appendix.**
(DOCX)

## Author Contributions

**Conceptualization:** Troy Puga, Yadi Liu, Peng Xiao, Ran Dai, Hongying Daisy Dai.

**Data curation:** Troy Puga, Yadi Liu, Hongying Daisy Dai.

**Formal analysis:** Yadi Liu, Peng Xiao, Ran Dai, Hongying Daisy Dai.

**Funding acquisition:** Hongying Daisy Dai.

**Project administration:** Troy Puga.

**Writing – original draft:** Troy Puga.

**Writing – review & editing:** Yadi Liu, Peng Xiao, Ran Dai, Hongying Daisy Dai.

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
