## [Decision Letter · Decision Letter 0]

12 Dec 2023

PONE-D-23-34516Genetic and Environmental Influence on Alcohol Intent and Alcohol Sips among U.S. ChildrenPLOS ONE

Dear Dr. Puga,

Thank you for submitting your manuscript to PLOS ONE. After careful consideration, we feel that it has merit but does not fully meet PLOS ONE’s publication criteria as it currently stands. Therefore, we invite you to submit a revised version of the manuscript that addresses the points raised during the review process. The revised manuscript should address all comments.

We look forward to receiving your revised manuscript.

Kind regards,

Petri Böckerman

Academic Editor

PLOS ONE

 [Research reported in this publication was supported by the National Institute on Drug Abuse under Award Number R21ES033066 (Dai).].  

Additional Editor Comments:

The revised manuscript should address all comments.

Reviewers' comments:

Reviewer's Responses to Questions

**Comments to the Author**

1. Is the manuscript technically sound, and do the data support the conclusions?

Reviewer #1: Partly

Reviewer #2: Yes

2. Has the statistical analysis been performed appropriately and rigorously? 

Reviewer #1: No

Reviewer #2: Yes

3. Have the authors made all data underlying the findings in their manuscript fully available?

Reviewer #1: No

Reviewer #2: Yes

4. Is the manuscript presented in an intelligible fashion and written in standard English?

Reviewer #1: No

Reviewer #2: Yes

5. Review Comments to the Author

Reviewer #1: Major comments:

1. In the declarations, the authors did not include an ethics statement (NA) while the study included human participants and should therefore include this statement.

2. The authors state that their data is publicly available from the National Data Archive, for a US researcher this might be sufficient information to find the data, but as a non-US researcher it does not. Moreover, the journal request including accession numbers which have not been provided.

3. Abstract: The abstract misses some important information, such as what are alcohol intents or alcohol sips, how are those constructs measured, why are they of import to study? I would also advise the authors to rephrase their study aim as only the context makes it clear they aim to study G and E of alcohol related traits; this is not explicitly mentioned.

a. In the abstract I could not find a definition of alcohol intent or alcohol sips, both constructs are again mentioned in the introduction [lines 112-113] without explanation.

4. On lines 197-202 the authors describe how the difference in MZ and DZ correlation can be used to determine what kind of twin model would best fit the data. Their description on the interpretation of the twin correlations is not correct. When rMZ is more than twice as large as rDZ we assume that both additive genetic (A) and common environmental (C) factors are involved, when this is less than twice as large (but not equally large) we assume that instead of common environmental factors dominant genetic factors (D) play a role, i.e., an ADE model and not an ACE model. Moreover, the authors do not report how the twin correlations were obtained, unless the Pearson correlations mentioned on line 203 refer to the twin correlations. In that case, I would recommend clarifying the method section, but I would also recommend reporting the intraclass correlations instead (these can easily be obtained from the mixed models).

5. Given the relatively low numbers of non-white individuals I wonder about the power to detect significant C or D effects. Adding power analyses for all subgroup analyses would clarify this point beyond the brief mention of possible power issues on lines 297-299.

6. Lines 509-512 (note for table 2): information on statistical methods is mentioned while this was not included in the method section. While a comparison of models is mentioned in the methods and in this table note, I cannot find the results for these comparisons, would it be possible to include these in a supplement? More concerning is the mention of ADCE models as it is not possible to assess both C and D in models using only MZ and DZ twin pairs.

Minor comments:

1. In general, I recommend the authors critically reread their manuscript for grammatical errors or wording that seems a little off. Please see a few of the more obvious things I (as a non-native speaker) picked up on in the first few pages:

a. line 50: “alcohol intents and alcohol sips is” should be “alcohol intents and alcohol sips are”

b. line 57 “proportion of the total alcohol phenotypic variation attributable to genes” and line 65 “demonstrated higher alcohol intent due to additive genes than Whites”; I struggle with attributable to genes/additive genes and would prefer (additive) genetics effects/factors.

c. lines 71-72: “alcohol susceptibility initiation” I think this should be “susceptibility to alcohol use initiation” or something similar?

d. line 86-87: “Youth males” is “young males?

e. lines 104-105: “Twin studies have shown that some degree of heritability may exist in those with alcohol addiction”, I can’t even precisely pinpoint what’s wrong with this sentence grammatically, but it needs work. Also “some degree of h2 may exist” is too vague, simply report what has been found.

f. Line 116: “current studies have yet to identify a single gene implicated in this” given that the authors (rightly) argue that alcohol use disorder is a polygenic disorder and cite some literature backing this, this sentence seems superfluous to me.

i. With regards to the literature cited, the authors state that research into sex or ancestry differences are scare, while true, they do not cite relevant research here.

g. Line 152: “twin status”, is an uncommon term in my opinion, and would recommend sticking to zygosity as the authors use throughout the remainder of the manuscript.

h. Line 167, not underlined while zygosity and sociodemographic factors were

i. Line 170: missing closing bracket after whiskey

j. Line 179, not on separate line while zygosity and alcohol intent and sips were

2. When including in text references sometimes authors include a space between a word and the reference and other times they don’t. PLOS ONE requires a space between the word and the references and has the numbered references in brackets, i.e., “The h2 of alcohol use disorder is XX [1].”

3. Luckily the number of twins excluded because their co-twin was not included in the study was very small, but for future studies I would not recommend excluding incomplete twin pairs, but use raw maximum likelihood estimates in structural equation analyses which allow for incomplete data (and is the most common statistical technique for twin modelling).

a. On a related note, it’s a shame more than 300 twins were excluded due to missing or inconsistent zygosity as assessed by questionnaires. Has this study also genotyped all twin individuals? And if so, would it not be preferable to obtain zygosity from DNA rather than survey (to ensure more pairs can be included, as the accuracy of survey-based zygosity compared to DNA-based is actually very good, see for example Ligthart et al. 2019 in TRHG)?

4. Lines 187 indicates both parents and children reported on perceived neighborhood safety, but if is unclear how the other sociodemographics were obtained: also from both parents and children? Or perhaps through record linkage of some sort? The main reason to ask for this clarification is that I feel it is unlikely that most children would know how much their parents earn and perhaps they may also not be fully aware of how much education their parents received.

5. The Rao-Scott Chi-square tests described in the first paragraph of the results (and related table 1) were not described in the methods section.

Reviewer #2: In this manuscript, the authors investigated the alcohol intent and alcohol sips among U.S. Children and analyzed different factors. The manuscript is very impressive and I do not have much to comment. However, the authors should add few lines in the discussion about how this study can help to mitigate the incidence of alcohol use disorder on the basis of the current findings.

6. PLOS authors have the option to publish the peer review history of their article (what does this mean?). If published, this will include your full peer review and any attached files.

Reviewer #1: No

Reviewer #2: **Yes: **Abhishek Basu

---

## [Author Response · Author response to Decision Letter 0]

9 Jan 2024

• We have reformatted the manuscript to meet PLOS One’s style requirements. 

 [Research reported in this publication was supported by the National Institute on Drug Abuse under Award Number R21ES033066 (Dai).]. 

• The following statement has been added in the manuscript “The funders had no role in study design, data collection and analysis, decision to publish, or preparation of the manuscript” See line 366-368.

• The following statement has been added to the cover letter “Research by H. Dai was partially supported by the National Institute on Drug Abuse under Award Numbers R21DA054818 and R01DA058992. The funding agency had no role in the design and conduct of the study; collection, management, analysis, and interpretation of the data; preparation, review, or approval of the manuscript; and decision to submit the manuscript for publication.”

• We have updated the data statement. The ABCD data can be found in the National Data Archive at the following DOI: 10.15154/z563-zd24. We have added the following statement in the cover letter “ All ABCD data study is publicly available through the National Data Archive at the following DOI: 10.15154/z563-zd24.”

• This data sharing statement is consistent with a recent publication [Reference: Zhou, Y., Pat, N., & Neale, M. C. (2023). Associations between resting state functional brain connectivity and childhood anhedonia: A reproduction and replication study. Plos one, 18(5), e0277158.]

• See Lines 406-408

• The title has been amended in the online submission to have an identical match the manuscript.

Additional Editor Comments:

The revised manuscript should address all comments.

• We have responded to all editor comments, reviewer comments, and reviewer questions.

Reviewers' comments:

• Thanks for your insightful suggestions and comments. We have addressed all reviewer’s responses to comments. See below.

Reviewer's Responses to Questions

• We have addressed all reviewer’s responses to questions. See below.

Comments to the Author

1. Is the manuscript technically sound, and do the data support the conclusions?

Reviewer #1: Partly

Reviewer #2: Yes

• We have clarified the definition of outcome variables, and addressed the comments point-by-point. 

2. Has the statistical analysis been performed appropriately and rigorously?

Reviewer #1: No

Reviewer #2: Yes

• We have clarified statistical method and addressed the comments point-by-point. 

3. Have the authors made all data underlying the findings in their manuscript fully available?

Reviewer #1: No

Reviewer #2: Yes

• We have updated the data availability statement in the online portal and the cover letter. The text now becomes “The ABCD study is publicly available under the National Institute of Mental Health National Data Archive (NDA) at https://nda.nih.gov/abcd. Interested researchers can sign the Data User Agreement with the NDA to access the ABCD study dataset.”

• This data sharing statement is consistent with a recent publication [Reference: Zhou, Y., Pat, N., & Neale, M. C. (2023). Associations between resting state functional brain connectivity and childhood anhedonia: A reproduction and replication study. Plos one, 18(5), e0277158.]

• See Lines 406-408.

4. Is the manuscript presented in an intelligible fashion and written in standard English?

Reviewer #1: No

Reviewer #2: Yes

• We have corrected any grammatical errors throughout the manuscript. The manuscript is presented in an intelligible fashion and written in standard English.

5. Review Comments to the Author

Reviewer #1: Major comments:

1. In the declarations, the authors did not include an ethics statement (NA) while the study included human participants and should therefore include this statement.

• The following statement has been added to the ethics statement declaration “The original ABCD study was approved by the University of California San Diego Institutional Review Board (IRB) and each local institutional IRB for the ABCD Study. The Institutional Review Board from the University of Nebraska Medical Center granted an exemption for the secondary data analysis of de-identified information from the ABCD study.” This can also be found in the methods section, see lines 152-156.

2. The authors state that their data is publicly available from the National Data Archive, for a US researcher this might be sufficient information to find the data, but as a non-US researcher it does not. Moreover, the journal request including accession numbers which have not been provided.

• We have provided the information regarding the data, including the DOI number. The ABCD study data can be found in the National Data Archive at the following DOI: 10.15154/z563-zd24. This information and use of the DOI is in compliance with journal requirements. 

• The text now becomes “The ABCD study is publicly available under the National Institute of Mental Health National Data Archive (NDA) at https://nda.nih.gov/abcd. Interested researchers can sign the Data User Agreement with the NDA to access the ABCD study dataset.”

• See Lines 406-408.

3. Abstract: The abstract misses some important information, such as what are alcohol intents or alcohol sips, how are those constructs measured, why are they of import to study? I would also advise the authors to rephrase their study aim as only the context makes it clear they aim to study G and E of alcohol related traits; this is not explicitly mentioned.

• Thanks for your suggestions. We have clarified the alcohol intents and alcohol sips in Abstract. See Lines 50-51. We also included the importance of this study in Abstract (see Lines 72-73) and clarified the study aim to dissect the additive genetic (A), common (C), and unique environmental (E) effects on alcohol traits (see Lines 56-59).

• Due to the space limit, we provided details about the measurement of those constructs in Measures section (see Lines 172-183) and the supplementary materials. 

a. In the abstract I could not find a definition of alcohol intent or alcohol sips, both constructs are again mentioned in the introduction [lines 112-113] without explanation.

• We regret not clarifying the definition of alcohol intent and alcohol sips before introducing them. In the revised manuscript, alcohol intent has been defined as “the susceptibility to initiating alcohol,” and alcohol sips as “the initiation of alcohol”. This has been clarified in the Abstract and the introduction. See lines 50-51 and 116.

• The details about the measurement of these constructs are provided in Measures section (see Lines 172-183). In the Supplementary document, we added the definition of variables and the URL of the code book. 

4. On lines 197-202 the authors describe how the difference in MZ and DZ correlation can be used to determine what kind of twin model would best fit the data. Their description on the interpretation of the twin correlations is not correct. When rMZ is more than twice as large as rDZ we assume that both additive genetic (A) and common environmental (C) factors are involved, when this is less than twice as large (but not equally large) we assume that instead of common environmental factors dominant genetic factors (D) play a role, i.e., an ADE model and not an ACE model. Moreover, the authors do not report how the twin correlations were obtained, unless the Pearson correlations mentioned on line 203 refer to the twin correlations. In that case, I would recommend clarifying the method section, but I would also recommend reporting the intraclass correlations instead (these can easily be obtained from the mixed models).

• In accordance with your suggestion, we have rewritten the genetic interpretation as “when the MZ correlation is larger than (e.g., 50%) the DZ correlation (30%), there is evidence of genetic influences, 2) when the MZ correlation (e.g., 42%) is similar to the DZ correlation (e.g., 42%), variance is likely attributed to environmental factors with little evidence of genetic influence, and 3) when the DZ correlation (60%) is larger than half of the MZ correlation (80%), variance is likely attributed to environmental factors.”

• Thanks for your suggestion, we also added ICC to Table 1 and the results are consistent with the Pearson correlation. 

• In the method section, we clarified that “ Intraclass correlation coefficients, alongside Pearson correlation coefficients and Bayesian information criterion (BIC) were provided.”

• See Lines 206-218 and 213-214.

5. Given the relatively low numbers of non-white individuals I wonder about the power to detect significant C or D effects. Adding power analyses for all subgroup analyses would clarify this point beyond the brief mention of possible power issues on lines 297-299.

• The statistical power analysis for the twin study was conducted by a prior study (Iacono et al, 2017). We have added results of power analysis from this study in the statistical method section “With 800 twin pairs recruited from four geographically diverse sites (Minnesota, Colorado, Virginia, and Missouri), the ABCD study offers the largest sample size to assess behavioral genetics among adolescents longitudinally. Because DZ twins are twice more prevalent than MZ pairs, the ABCD twin study oversampled MZ twins to balance the zygosity distribution. The power analysis considered the impact of sample size and measurement type on detecting heritable variation. With the proposed sample, this study achieved a power greater than 0.8 to detect additive genetic effects when the heritable variation exceeds 0.3 for continuous measures. The power of binary measures depends on the prevalence of outcomes, while ordinal measures provide more power than binary measures. Despite practical limitations, the study was designed to maximize statistical power.”

• Furthermore, we added a limitation stating “while the ABCD study has one of the largest twin hub data for longitudinal assessment of behavioral genetics among adolescents, its capacity to detect racial disparities might be limited in the ACE model due to the relatively small sample sizes within certain racial minority groups (e.g., Blacks ad Hispanics). Therefore, caution should be exercised when generalizing findings to these populations, and additional research with larger samples from diverse racial backgrounds may be necessary to fully understand the implications for different demographic groups.”

• See Lines 218-228 and 368-377.

6. Lines 509-512 (note for table 2): information on statistical methods is mentioned while this was not included in the method section. While a comparison of models is mentioned in the methods and in this table note, I cannot find the results for these comparisons, would it be possible to include these in a supplement? More concerning is the mention of ADCE models as it is not possible to assess both C and D in models using only MZ and DZ twin pairs.

• We agree and apologize for citing the ADCE model as a typo.

• The revised text now becomes “Due to the identifiability problem, we did not dissect additive genetic effects, dominance genetic effects, and shared environmental effects.” 

• See Lines 215-216.

Minor comments:

1. In general, I recommend the authors critically reread their manuscript for grammatical errors or wording that seems a little off. Please see a few of the more obvious things I (as a non-native speaker) picked up on in the first few pages:

• Thanks for your review and suggestions. We have proofread the manuscript and have corrected any grammatical errors. See the responses below.

a. line 50: “alcohol intents and alcohol sips is” should be “alcohol intents and alcohol sips are”

• We have changed “alcohol intents and alcohol sips is” to “alcohol intents and alcohol sips are”. See line 51.

b. line 57 “proportion of the total alcohol phenotypic variation attributable to genes” and line 65 “demonstrated higher alcohol intent due to additive genes than Whites”; I struggle with attributable to genes/additive genes and would prefer (additive) genetics effects/factors.

• “Proportion of the total alcohol phenotypic variation attributable to genes” has been changed to the proportion of the total alcohol phenotypic variation attributable to additive genetic factors” to improve clarity and readability. See line 58

• “Demonstrated higher alcohol intent due to additive genes than Whites” has been changed to “demonstrated higher alcohol intent due to additive genetic factors than Whites” to improve clarity and readability. See line 66.

c. lines 71-72: “alcohol susceptibility initiation” I think this should be “susceptibility to alcohol use initiation” or something similar?

• “alcohol susceptibility initiation” has been changed to “susceptibility to alcohol initiation”. See line 72-73.

d. line 86-87: “Youth males” is “young males?

• “Youth males” has been changed to “Young males”. See line 88.

e. lines 104-105: “Twin studies have shown that some degree of heritability may exist in those with alcohol addiction”, I can’t even precisely pinpoint what’s wrong with this sentence grammatically, but it needs work. Also “some degree of h2 may exist” is too vague, simply report what has been found.

• “Twin studies have shown that some degree of heritability may exist in those with alcohol addiction” has been changed to “Previous twin studies have shown that alcohol dependence is heritable” for improved readability and a clearer report of findings. 

• See Lines 106.

f. Line 116: “current studies have yet to identify a single gene implicated in this” given that the authors (rightly) argue that alcohol use disorder is a polygenic disorder and cite some literature backing this, this sentence seems superfluous to me.

• “Current studies have yet to identify a single gene implicated in this” has been removed as it is not necessary.

i. With regards to the literature cited, the authors state that research into sex or ancestry differences are scare, while true, they do not cite relevant research here.

• The following citations of recent literature have been added to indicate the scarcity of information that exists with regard to genetic understanding of sex or ancestry differences. See line 122-123.

• Edenberg HJ, Gelernter J, Agrawal A. Genetics of Alcoholism. Curr Psychiatry Rep. 2019;21(4):26. Published 2019 Mar 9. doi:10.1007/s11920-019-1008-1

• Hitzemann R, Bergeson SE, Berman AE, et al. Sex Differences in the Brain Transcriptome Related to Alcohol Effects and Alcohol Use Disorder. Biol Psychiatry. 2022;91(1):43-52. doi:10.1016/j.biopsych.2021.04.016

g. Line 152: “twin status”, is an uncommon term in my opinion, and would recommend sticking to zygosity as the authors use throughout the remainder of the manuscript.

• “Twin status” has been changed to “Zygosity” for consistency. See line 157.

h. Line 167, not underlined while zygosity and sociodemographic factors were

• Alcohol intent and Sips are underlined to be consistent with the underlining of zygosity and sociodemographic factors. See line 172.

i. Line 170: missing closing bracket after whiskey

• A closing bracket has been added after whiskey. See line 175.

j. Line 179, not on separate line while zygosity and alcohol intent and sips were

• Sociodemographic factors have been placed on a separate line. See lines 184-185.

2. When including in text references sometimes authors include a space between a word and the reference and other times they don’t. PLOS ONE requires a space between the word and the references and has the numbered references in brackets, i.e., “The h2 of alcohol use disorder is XX [1].”

• We have reformatted references using the PLOS One style. 

3. Luckily the number of twins excluded because their co-twin was not included in the study was very small, but for future studies I would not recommend excluding incomplete twin pairs, but use raw maximum likelihood estimates in structural equation analyses which allow for incomplete data (and is the most common statistical technique for twin modelling).

• We included all twin pairs in the analysis, except for cases where there were errors or inconsistencies in the recorded zygosity information. We chose to exclude these cases due to concerns about the accuracy of the data entry.

• We agree with your suggestion and have added a limitation statting: “Fourth, this study excluded cases where there were errors or inconsistencies in the recorded zygosity information due to concerns about the accuracy of the data entry. Future research should investigate using raw maximum likelihood estimates in structural equation modeling, as this method can accommodate incomplete data.”

• See Lines 370-374.

a. On a related note, it’s a shame more than 300 twins were excluded due to missing or inconsistent zygosity as assessed by questionnaires. Has this study also genotyped all twin individuals? And if so, would it not be preferable to obtain zygosity from DNA rather than survey (to ensure more pairs can be included, as the accuracy of survey-based zygosity compared to DNA-based is actually very good, see for example Ligthart et al. 2019 in TRHG)?

• You are correct, and we follow the same process to determine the zygosity. In other words, genetic inference was initially used to determine their zygosity status, and additional information (e.g., physical characteristics) was used for those individuals with missing zygosity status. We excluded 169 pairs of twins (n=336) due to a lack of zygosity information from DNA and they also can not be inferred from other information. 

• We clarified in the method section that “For twins, genetic inference was initially used to determine their zygosity status, categorizing them as either MZ or DZ. In instances where zygosity status was unavailable, researchers resorted to physical characteristics, such as facial appearance, complexion, hair features (color, texture, curliness, pattern, amount, and darkness), ear appearance, hair type, and eye color, to differentiate between MZ and DZ twins.”

• See Lines 163-168.

4. Lines 187 indicates both parents and children reported on perceived neighborhood safety, but if is unclear how the other sociodemographics were obtained: also from both parents and children? Or perhaps through record linkage of some sort? The main reason to ask for this clarification is that I feel it is unlikely that most children would know how much their parents earn and perhaps they may also not be fully aware of how much education their parents received.

• We regret not clarifying the information in the original manuscript. Parents provided baseline demographic data through the PhenX Demographics survey. See ABCD Parent Protocol Summary: Baseline (https://abcdstudy.org/wp-content/uploads/2019/12/flyer_protocol_Baseline_new.pdf)

• A statement has been provided to clarify that parents provided the sociodemographic data. See line 185.

5. The Rao-Scott Chi-square tests described in the first paragraph of the results (and related table 1) were not described in the methods section.

• The following statement has been added to the statistical methods section: “Sociodemographic information was collected and compiled. Rao-Scott Chi-Square tests were conducted to detect sociodemographic differences between the intent and sip groups. See Table 1.”

Reviewer #2: In this manuscript, the authors investigated the alcohol intent and alcohol sips among U.S. Children and analyzed different factors. The manuscript is very impressive and I do not have much to comment. However, the authors should add few lines in the discussion about how this study can help to mitigate the incidence of alcohol use disorder on the basis of the current findings.

• Thanks for your comments. In accordance with your suggestion, the following has been added to the discussion in order to illuminate how these findings can help mitigate the incidence of alcohol use disorder “The findings in this study will be valuable for informing future public health interventions in reducing early-age alcohol initiation and alcohol use disorder. The knowledge gained from this study will allow public health practitioners to apply health promotion theories and interventions to individuals who may be at higher risk for alcohol intent and initiation. Public health interventions that use the information from this study will have the potential to reduce early-age alcohol use in the high-risk populations.” 

• See line 354-359.

6. PLOS authors have the option to publish the peer review history of their article (what does this mean?). If published, this will include your full peer review and any attached files.

Do you want your identity to be public for this peer review? For information about this choice, including consent withdrawal, please see our Privacy Policy.

Reviewer #1: No

Reviewer #2: Yes: Abhishek Basu

• Thank you for your insightful comments and suggestions, which have significantly improved this manuscript.

---

## [Decision Letter · Decision Letter 1]

25 Jan 2024

Genetic and Environmental Influence on Alcohol Intent and Alcohol Sips among U.S. Children-Effects across Sex, Race, and Ethnicity

PONE-D-23-34516R1

Dear Dr. Puga,

We’re pleased to inform you that your manuscript has been judged scientifically suitable for publication and will be formally accepted for publication once it meets all outstanding technical requirements.

Kind regards,

Petri Böckerman

Academic Editor

PLOS ONE

Additional Editor Comments (optional):

I am happy with the revised version of the paper.

Reviewers' comments:

Reviewer's Responses to Questions

**Comments to the Author**

1. If the authors have adequately addressed your comments raised in a previous round of review and you feel that this manuscript is now acceptable for publication, you may indicate that here to bypass the “Comments to the Author” section, enter your conflict of interest statement in the “Confidential to Editor” section, and submit your "Accept" recommendation.

Reviewer #1: All comments have been addressed

Reviewer #2: All comments have been addressed

2. Is the manuscript technically sound, and do the data support the conclusions?

Reviewer #1: (No Response)

Reviewer #2: Yes

3. Has the statistical analysis been performed appropriately and rigorously? 

Reviewer #1: (No Response)

Reviewer #2: Yes

4. Have the authors made all data underlying the findings in their manuscript fully available?

Reviewer #1: (No Response)

Reviewer #2: Yes

5. Is the manuscript presented in an intelligible fashion and written in standard English?

Reviewer #1: (No Response)

Reviewer #2: Yes

6. Review Comments to the Author

Reviewer #1: (No Response)

Reviewer #2: The authors have responded well to the reviewers' comments and no further comments are necessary for this manuscript.

7. PLOS authors have the option to publish the peer review history of their article (what does this mean?). If published, this will include your full peer review and any attached files.

Reviewer #1: No

Reviewer #2: **Yes: **Abhishek Basu

---

## [Editor Report · Acceptance letter]

6 Feb 2024

PONE-D-23-34516R1 

PLOS ONE

Dear Dr. Puga, 

I'm pleased to inform you that your manuscript has been deemed suitable for publication in PLOS ONE. Congratulations! Your manuscript is now being handed over to our production team.

Kind regards, 

on behalf of

Professor Petri Böckerman 

Academic Editor

PLOS ONE